# Harnessing Nature’s Defence: The Antimicrobial Efficacy of Pasteurised Cattle Milk-Derived Extracellular Vesicles on *Staphylococcus aureus* ATCC 25923

**DOI:** 10.3390/ijms25094759

**Published:** 2024-04-26

**Authors:** Dulmini Nanayakkara Sapugahawatte, Kasun Godakumara, Mihkel Mäesaar, Gayandi Ekanayake, Getnet Balcha Midekessa, Madhusha Prasadani, Suranga Kodithuwakku, Mati Roasto, Aneta Andronowska, Alireza Fazeli

**Affiliations:** 1Institute of Veterinary Medicine and Animal Sciences, Estonian University of Life Sciences, Fr. R. Kreutzwaldi 62, 51006 Tartu, Estonia; dulmini@emu.ee (D.N.S.); kasun.godagedara@emu.ee (K.G.); gayandi.mudiyanselage@emu.ee (G.E.); getnet.balcha@ut.ee (G.B.M.); madhusha.gamage@emu.ee (M.P.); suranga.kodithuwakku@emu.ee (S.K.); 2Chair of Veterinary Biomedicine and Food Hygiene, Estonian University of Life Sciences, Fr. R. Kreutzwaldi 56/3, 51006 Tartu, Estonia; mihkel.maesaar@emu.ee (M.M.); mati.roasto@emu.ee (M.R.); 3Department of Pathophysiology, Institute of Biomedicine and Translational Medicine, University of Tartu, Ravila 14b, 50411 Tartu, Estonia; 4Department of Animal Sciences, Faculty of Agriculture, University of Peradeniya, Peradeniya 20400, Sri Lanka; 5Institute of Animal Reproduction and Food Research, Polish Academy of Sciences, Juliana Tuwima St. 10, 10-748 Olsztyn, Poland; a.andronowska@pan.olsztyn.pl; 6Division of Clinical Medicine, School of Medicine & Population Health, University of Sheffield, Level 4, Jessop Wing, Tree Root Walk, Sheffield S10 2SF, UK

**Keywords:** *Staphylococcus aureus*, extracellular vesicles, novel antimicrobials, pasteurised milk, antibacterial activity, dietary EVs

## Abstract

Increasing antimicrobial resistance (AMR) challenges conventional antibiotics, prompting the search for alternatives. Extracellular vesicles (EVs) from pasteurised cattle milk offer promise, due to their unique properties. This study investigates their efficacy against five pathogenic bacteria, including *Staphylococcus aureus* ATCC 25923, aiming to combat AMR and to develop new therapies. EVs were characterised and tested using various methods. Co-culture experiments with *S. aureus* showed significant growth inhibition, with colony-forming units decreasing from 2.4 × 10^5^ CFU/mL (single dose) to 7.4 × 10^4^ CFU/mL (triple doses) after 12 h. Milk EVs extended lag time (6 to 9 h) and increased generation time (2.8 to 4.8 h) dose-dependently, compared to controls. In conclusion, milk EVs exhibit dose-dependent inhibition against *S. aureus*, prolonging lag and generation times. Despite limitations, this suggests their potential in addressing AMR.

## 1. Introduction

In recent years, global antimicrobial resistance (AMR) has emerged as a significant challenge, posing a serious obstacle to our ability to effectively combat bacterial infections in both animal and human populations. In 2019, 1.27 million deaths were directly attributed to drug-resistant infections worldwide [1,2]. Looking ahead to 2050, a more alarming scenario may apply, with estimates suggesting that up to 10 million deaths annually could be associated with AMR, if prompt and decisive action is not taken [1]. The upcoming crisis emphasises the urgent need for innovative solutions. Additionally, the worrisome limitations in the development of novel antimicrobial drugs compound the challenge, endangering our capacity to address the dynamic landscape of infectious diseases on a global scale [3].

As conventional antibiotics encounter a diminishing efficacy against resistant pathogens, the imperative for alternative antimicrobial strategies intensifies [4]. Extracellular vesicles (EVs), naturally occurring membrane-bound particles, have emerged as promising candidates in this pursuit, due to their unique characteristics and biogenesis [5].

EVs are heterogeneous lipid bilayer-enclosed structures released by various cell types, including bacteria, archaea, and eukaryotic cells [6]. They play a crucial role in intercellular communication and the transport of bioactive molecules. These vesicles encompass exosomes, microvesicles, and apoptotic bodies, each distinguished by size, biogenesis pathways, and cargo composition [6].

Initially regarded as cell membrane fragments for waste discharge, EVs have since been recognised for their involvement in various physiological processes, including angiogenesis, inflammation, immune response, and neuron signalling. Mammalian milk, notably rich in EVs, contains diverse RNA, lipids, and proteins within its EVs [5,6]. The bilayer membrane structure of milk-derived EVs enables them to withstand gastric and pancreatic digestion, facilitating absorption by intestinal cells and subsequent biological activities. Furthermore, EVs can traverse biological barriers and deliver cargo to target cells, making them attractive candidates for antimicrobial applications [7,8]. 

These EVs could exhibit antimicrobial properties, contributing to host defence against infections. These properties might be due to several mechanisms. First, EVs can carry antimicrobial molecules such as peptides, microRNAs, and enzymes that directly inhibit pathogen growth or virulence [9]. Second, EVs modulate the immune response by delivering signalling molecules that activate innate immune pathways or recruit immune cells to combat infections [10]. Third, they compete with pathogen-derived vesicles for host cell binding sites, hindering pathogen entry and establishment of infection [11]. Fourth, EVs disrupt microbial biofilms, rendering pathogens more susceptible to antimicrobial agents and immune clearance [12]. Lastly, EVs can interfere with microbial quorum-sensing systems, regulating virulence factor production and biofilm formation [13]. These insights highlight the multifaceted role of EVs in both microbial communication and host defence, enriching our understanding of their biological significance. Thus, harnessing the potential of EVs as natural vehicles for therapeutic cargo delivery represents a promising avenue in the ongoing quest to address the challenges posed by AMR.

Milk EVs have recently gained significant attention, primarily owing to their intricate bioactive characteristics [7]. These vesicles, originating from the mammary glands of lactating animals, carry a diverse cargo of proteins, lipids, and nucleic acids, reflecting the complexity of their cellular origin. Milk EVs are implicated in various physiological processes, including immune modulation, nutrient transport, and cellular communication [14]. Moreover, the stability and bioavailability of these vesicles make them particularly interesting for therapeutic applications, prompting a focused exploration into their potential antimicrobial properties [15].

This study embarks on a targeted investigation into pasteurised cattle milk EVs, aiming to uncover their efficacy against common bacterial pathogens including *Staphylococcus aureus* ATCC 25923—an essential bacterial pathogen responsible for a spectrum of infections both in humans and animals. *S. aureus* poses a significant public health concern, due to its ability to cause a range of illnesses, from common skin and soft tissue ailments to severe systemic diseases and food poisoning [16,17]. The bacterium’s resilience and evolving resistance to conventional antibiotics underscore the urgency of exploring alternative antimicrobial strategies [18]. By investigating the antimicrobial potential of pasteurised cattle milk EVs, this research seeks to contribute valuable insights into the development of novel therapeutic interventions that could address the challenges posed by *S. aureus* infections.

This study hypothesises that milk-derived EVs inherently possess properties that can serve as an alternative strategy to combat bacterial infections, contributing to the overall mitigation of AMR. The primary aim is to comprehensively assess the antimicrobial efficacy of these milk EVs, while the specific objectives include elucidating their effectiveness against *S. aureus*, addressing the global need for alternative antimicrobial solutions, and contributing insights to innovative approaches in countering the challenges posed by AMR in the coming decades. Through this research, we aim to provide insights into the development of novel therapeutic interventions, with the potential to support our defences against the evolving scenario of *S. aureus* infections.

## 2. Results

### 2.1. Isolation and Characterisation of Pasteurised Milk EVs

EVs were isolated from commercially available pasteurised cow’s milk. Transmission electron microscopy (TEM) analysis revealed the size and morphology of milk EVs. Notably, TEM images showed that the diameter of the EVs ranged from approximately 100 to 200 nm, while also displaying an imperfectly spherical or cup shape (Figure 1A).

Nanoparticle Tracking Analysis (NTA) was used to determine the size distribution of the EVs isolated from commercially available pasteurised milk samples. The calculated average sizes of the isolated EVs were ~170 nm (Figure 1B). 

The presence of CD63 and CD81 in Western blotting indicates the presence of EVs in pasteurised cow’s milk (Figure 1C). Overall, these results indicate that pasteurised cow’s milk EVs were successfully isolated using tangential flow filtration (TFF).

### 2.2. Milk EV Activity on the Bacterial Growth Curve

In a co-culture of bacteria and milk EVs, the effect of EVs on the growth of various bacterial strains was investigated and the maximum percentage growth difference was compared to the control at six different time points of incubation, as enumerated in Table 1. The results showed noticeable reductions in the growth of *S. aureus* ATCC 25923, *B. subtilis* ATCC 6633, and *B. cereus* ATCC 11778 at 9 h of incubation, while a growth promotion in *P. aeruginosa* ATCC 27853 and Escherichia coli ATCC 53868 was observed in the presence of milk EVs (Table 1; Figure 2). The most substantial growth inhibition occurred in *S. aureus* at the 9 h mark and reached statistical significance (*p* = 0.0222), based on *t*-test analysis. Importantly, the growth of *E. coli* was the least affected strain in the presence of milk EVs, when compared to control conditions (Figure 2).

These results suggest that the impact of milk EVs on bacterial growth varies among different strains, highlighting the strain-specific responses in co-culture settings.

### 2.3. Periodic Dosing of Milk EVs on S. aureus Growth and CFU Reduction Assay

Concerning the preliminary results in Figure 2, *S. aureus* ATCC 25923 was further tested for its activity in the presence of three consecutive milk EV doses. *S. aureus* ATCC 25923, cultured in the planktonic phase at a concentration of 1 × 10^7^ CFU/mL, was co-cultured with pasteurised milk EVs at a concentration of 1 × 10^9^ EVs/mL in Muller–Hinton broth. Two subsequent doses of EVs were administered at 3 h and 6 h time points and incubation continued for 24 h at 35 °C. The co-cultivation of *S. aureus* with EVs demonstrated a distinct ability to delay bacterial growth in a dose-dependent manner (Figure 3). This observation revealed the potential inhibitory impact of pasteurised milk EVs on the growth dynamics of *S. aureus*, highlighting their dose-responsive behaviour in this co-culture system.

The CFU/mL decreased from 2.4 × 10^5^ (single dose) to 2.3 ×10^5^ (double doses) and further to 7.4 × 10^4^ (triple doses), after 12 h of incubation (Figure 3A). Furthermore, percentage inhibition was 33.3%, 11.7%, and 47.4%, when EV doses increased from one to three after 12 h of incubation (Figure 3B). 

The CFU/mL decreased from 2.4 × 10^5^ (single dose) to 2.3 × 10^5^ (double doses) and further to 7.4 × 10^4^ (triple doses), after 12 h of incubation (Figure 4A). Furthermore, percentage inhibition was 33.3%, 11.7%, and 47.4%, when EV doses increased from one to three after 12 h of incubation (Figure 4B).

### 2.4. Prediction of Bacterial Growth-Related Parameters

Lag time and generation time were calculated using a model-free system, by identifying features of the density data and its derivatives. We observed that milk EVs can extend the lag time of *S. aureus* ATCC 25923, increasing it from 6 h (single dose, *p* = 0.0208; double doses, *p* = 0.0202) to 9 h (triple doses, *p* = 0.0109), compared to the control’s 4 h (Figure 5A). Periodic milk EV dosing also prolonged the overall generation time of this bacterial strain, from 2.8 h (single dose) to 3.5 h (double doses) and 4.8 h (triple doses), compared to the control’s 3.4 h (Figure 5B).

## 3. Discussion

Global AMR is an urgent and severe public health threat, with projections indicating a potential annual toll of up to 10 million lives by 2050 [2,3,19]. This upcoming situation highlights the crucial requirement for new and innovative ways to fight microbes, especially considering the challenges in creating new antibiotics [20]. Conventional antibiotic efficacy is weakening against increasingly resistant pathogens, necessitating a swift and transformative response to safeguard public health [21]. It is essential to prioritise and adopt novel approaches to address the growing impact of AMR and also to address our ability to effectively combat bacterial infections.

EVs have gained growing interest as promising therapeutic agents, owing to their distinctive characteristics [22]. Functioning as natural vehicles for intercellular communication and cargo delivery, EVs play a pivotal role in modulating recipient cell functions through the transfer of bioactive molecules such as proteins, lipids, nucleic acids, and metabolites [23,24]. Moreover, recent research has demonstrated that bacteria can uptake EVs and can alter bacterial gene expression [25]. EVs’ natural capability makes them potential contributors to new ways of fighting microbes, especially given the decreasing effectiveness of regular antibiotics against pathogens. Their unique properties, including the capacity to navigate biological barriers [26] and deliver cargo to target cells, make EVs an attractive candidate for addressing the challenges posed by antimicrobial resistance. Recognising the multifaceted role of EVs in intercellular signalling and their potential in combatting pathogenic bacteria underscores the significance of exploring these naturally occurring vesicles as a novel frontier in the search for effective antimicrobial solutions.

Compared to milk protein, fat, and hormones, milk-derived EVs are less frequently studied components of milk and the use of milk EVs, either as stand-alone drugs, drug carriers, or functional dietary components, has often been suggested in recent years. In the current study, TEM provided high-resolution images, revealing the consistent size and structure of EVs, with diameters ranging from approximately 100 to 200 nm and displaying an imperfectly spherical or cup shape, which resembles the results of previous reports [16]. Furthermore, NTA complemented this by determining the size distribution, with an average size of 170 nm, which is also within the previously reported range of 32.7 to 255 nm [25]. The International Society for Extracellular Vesicles (ISEVs) has indicated that EVs can be characterised using Western blotting, to identify membrane trafficking proteins and transmembrane proteins [27]. In the current study, Western blotting confirmed the presence of essential EV markers, CD63 and CD81, in milk EVs, demonstrating their identity and suggesting that the protein compositions of mammalian EVs are alike [25]. These comprehensive methods collectively affirmed the reliability of milk EV enrichment using the TFF method, highlighting the uniformity in size, structural integrity, and the presence of key molecular markers crucial for their functional identity.

In the bacterial growth inhibition assays, the co-culture experiments revealed diverse effects on different bacterial strains when exposed to milk-derived EVs. Notably, distinct strain-specific responses were observed, with varying degrees of growth inhibition and/or promotion. A study conducted by Yu et al. reported that milk exosomes could promote the growth of *Escherichia coli* K-12 MG1655 and *Lactobacillus plantarum* WCFS1 [25]; whereas, in our study, we also observed a slight growth promotion in *E. coli* ATCC 53868. Furthermore, an in vivo study explained that milk EVs could alter the gut microbiota composition by increasing *Clostridiaceae*, *Ruminococcaceae*, and *Lachnospiraceae*, while decreasing the relative abundance of *Bacteroidales* and *Helicobacteraceae*. This resulted in modulating their metabolites and increased the abundance of short-chain fatty acids [28]. On the contrary, *S. aureus*, in the current study, exhibited a statistically significant percentage growth inhibition, which can also be explained as a bacteriostatic behaviour, emphasising the efficacy of milk EVs’ activity against pathogenic bacteria. The varied responses among bacterial strains highlight the intricate interactions between EVs and different microbes, suggesting a potential specificity in the antimicrobial activity of these vesicles. The statistically significant growth inhibition of *S. aureus* ATCC 25923 further accentuates the potential of milk EVs as targeted antimicrobial agents with modifications, emphasising the importance of understanding strain-specific responses in the development of novel therapeutic interventions against bacterial infections. 

The periodic dosing experiments with milk EVs demonstrated a compelling dose-dependent inhibition of *S. aureus* growth. As the dosage of EVs increased, a noticeable delay in bacterial growth, extended lag time, and prolonged generation time were consistently observed in the current study. This indicates that the bacteriostatic effect of milk EVs on *S. aureus* is dose-responsive and impacts multiple facets of the bacterial growth dynamics. Similarly, Tong et al. reported that there was a significantly different improvement in healthy microbiota with the increase in EVs in cecum in mouse models, compared to EV-untreated controls [28]. The observed delay in growth and altered growth kinetics with increasing EV doses emphasises the intricate relationship between EV concentration and its inhibitory effects. These findings highlight the potential of milk EVs as effective modulators of bacterial growth, suggesting their ability to control the growth of *S. aureus* in a dose-dependent manner in both the optical density screening method and CFU/mL reduction assay.

The model-based predictions of bacterial growth-related parameters revealed significant insights into the dynamics of milk EV-mediated inhibition. The extension of lag time, from 6 h with a single dose to 9 h with triple doses, signifies a dosage-dependent delay in the initiation of bacterial growth. Concurrently, the increased generation time, progressing from 2.8 h (single dose) to 4.8 h (triple doses), emphasises the impact of EVs on decreasing the rate of bacterial growth. These predictions are crucial in understanding the bacterial dynamics in the presence of milk EVs and they also provide a quantitative framework on how different dosages of EVs influence the key parameters governing bacterial growth [29,30]. The model-based approach enhances our ability to anticipate and interpret the temporal dynamics of EV-mediated effects, contributing valuable insights into the potential mechanisms underlying their antimicrobial efficacy against *S. aureus*.

The findings of this study hold significant implications for the development of alternative antimicrobial strategies, in the face of escalating antimicrobial resistance. The demonstrated efficacy of milk EVs in inhibiting the growth of *S. aureus* ATCC 25923 emphasises their potential use as valuable possible antibacterial agents against essential pathogenic bacteria. The dose-dependent nature of EV-mediated inhibition and the associated alterations in bacterial growth kinetics open avenues for the precise modulation of antimicrobial effects. Future directions could involve further investigation into the specific mechanisms of action underlying the antimicrobial properties of milk-derived EVs. Exploring how EVs interact with bacterial membranes, influence gene expression, or modulate immune responses would enhance our understanding of their therapeutic potential. Additionally, investigating the efficacy of milk-derived EVs in more complex biological systems, such as animal models followed by clinical settings, would provide crucial insights into their translational potential. These avenues of research hold promise for the continued development of innovative and effective antimicrobial interventions, addressing the urgent global challenge of antimicrobial resistance.

Milk-derived EVs seem to offer targeted antimicrobial action, halting harmful bacteria, while preserving beneficial microbiota crucial for health. This precision minimises collateral damage seen with broad-spectrum antibiotics, reducing microbial imbalance. Milk EVs also show promise in combatting antimicrobial resistance by selectively pressuring harmful bacteria, potentially minimising resistance development, according to the current research, and their specificity represents a paradigm shift, offering effective microbial control, while safeguarding microbiota balance, and promising transformative antimicrobial strategies for the future. 

This study, while providing valuable insights, is not without its limitations and challenges. Firstly, the need for further in-depth mechanistic studies arises to elucidate the specific molecular pathways through which milk EVs exert their antimicrobial effects. Understanding the precise mechanisms of action would enhance the scientific community’s grasp of the potential therapeutic applications of these vesicles. Additionally, existing data have demonstrated the antimicrobial activity of certain antibacterial peptides found in bovine milk, such as Bovine k-casein, Kappacin A, and Isracidin, particularly against Gram-positive bacteria. Consequently, it is plausible that these antimicrobial peptides could be enriched within EVs, thereby conferring visible antimicrobial effects [31]. Incorporating these findings into our mechanistic studies could provide a more comprehensive understanding of the antimicrobial properties of milk EVs. Moreover, incorporating milk samples from diverse sources in future research could enhance our understanding of milk-derived EV activity against pathogens. 

Additionally, the inherent heterogeneity of EVs, both in terms of composition and cargo, poses a challenge. Variations in enrichment methods and EV composition could influence their antimicrobial efficacy, necessitating comprehensive analyses to capture this complexity. Moreover, this study focused on *S. aureus* ATCC 25923 and extrapolating the findings to other bacterial species requires cautious consideration of potential variations in responses. Addressing these limitations would contribute to a more robust understanding of the broader applications and mechanisms of milk EVs in antimicrobial interventions.

## 4. Materials and Methods

This study was carried out based on the following two main experimental setups, as explained in Figure 6:

### 4.1. Preparation, Enrichment, and Characterisation of Milk EVs

#### 4.1.1. Extracellular Vesicle Enrichment from Pasteurised Milk

Commercially available pasteurised low-fat milk (*n* = 3) was used as the source material to produce milk EVs (1.8% Tere joogipiim, TERE AS, Lelle 22, Tallinn 11318, Estonia). Glacial acetic acid (036289.K3, Thermo fisher Scientific, Waltham, MA, USA) was used to decrease the pH of the milk to 4.6, causing the milk to curdle and separate the whey liquid. Milk was left in +4 °C to curdle for 1 h and the curds were separated using Whatman grade 4 qualitative filter papers (1004-150, Cytiva, Marlborough, MA, USA) and 0.45 µm bottle top dead-end filters (CLS430512, Corning^®^, Sigma Aldrich, Burlington, MA, USA). Collected filtrates were subjected to TFF using a benchtop TFF system (Centramate, Cytiva, MA, USA), with a 300 kDa molecular weight cutoff polyethersulfone membrane filter (Omega PES membrane, 0.02 m^2^ surface area, Cytiva, MA, USA), run in a closed circle. The retentate was allowed to concentrate until the required enrichment was achieved. The filtrate was concentrated and the buffer was replaced with Dulbecco’s PBS (Sigma-Aldrich Co., St. Louis, MO, USA). The harvested EVs were centrifugated at 3000× *g* and the filtrate was stored at −80 °C until further use.

#### 4.1.2. Nanoparticle Tracking Analysis (NTA)

NTA measurements of milk EVs were carried out using a ZetaView^®^ PMX 420 QUATT instrument (Particle Metrix GmbH, Inning am Ammersee, Germany) coupled with the software version 8.05.14 SP7. NTA’s laser and microscope were auto-aligned using 100 nm polystyrene standard beads (Applied Microspheres B.V., Leusden, The Netherlands). The standards were suspended in particle-free Milli-Q water, whereas the EV samples were diluted in PBS for analyses. The particle size distribution and number were counted at 11 frames per cycle, under a sensitivity of 72 and a shutter value of 100 [32].

#### 4.1.3. Transmission Electron Microscopy (TEM)

TEM analysis was performed on pooled EVs obtained from milk samples, as described previously [31]. Formvar/carbon-coated 200 mesh grids (Agar Scientific Ltd., tansted, Essex, UK) were placed on 20 μL droplets of purified milk EVs for 20 min and the droplets were allowed to be absorbed on the grid. Then, the grids were incubated with 2% uranyl acetate (Polysciences, Touhy Avenue Niles, IL, USA) for 5 min and air-dried to obtain contrasted images of EVs. The EVs were visualised using JEM 1400 TEM (JEOL Ltd., Tokyo, Japan) with a numeric camara (Morada TEM CCD camera (Olympus, Germany) at 80 kV. Finally, the digital images of EVs were captured using a numeric camera (Morada TEM CCD camera, Olympus).

#### 4.1.4. SDS–PAGE and Western Blotting

Western blot analysis was performed on the milk EVs enriched with TFF and whey sample, according to the method described elsewhere [32]. Briefly, EVs were lysed and proteins were extracted using RIPA buffer (Thermo Fisher Scientific, Waltham, MA, USA), with a protease inhibitor cocktail (cat. 535140, EMD Millipore Corp, Burlington, MA, USA). Briefly, the sample and buffer solution were mixed in a 1:1 ratio and were incubated on ice for 15 min, followed by centrifugation at 15,000× *g* for 5 min at 4 °C. The supernatant was collected and the protein concentration was measured using the Pierce BCA Protein Assay Kit (cat. 23250, Thermo Fisher Scientific), according to the manufacturer’s instructions. The protein concentrations of all the samples were calculated and normalised to the total protein content. Finally, the samples were denatured at 95 °C for 5 min in loading buffer with β-mercaptoethanol for CD81 and non-reducing Laemmli buffer for CD63. 

The proteins were separated using 12% SDS–PAGE and were transferred to the PVDF membrane, followed by blocking in 5% non-fat milk for 1 h at room temperature. The membrane was incubated with the primary anti-CD63 antibody (cat. no MCA2042GA, 1:1000, Bio-Rad, Hercules, CA, USA) and anti-CD81 antibody (cat. no SC-166029, 1:1000, Santa Cruz Biotechnology, Dalla, TX, USA) overnight at 4 °C, followed by incubation with horseradish peroxidase-conjugated goat anti-mouse secondary antibody (cat. no G21040, 1:10,000, ThermoFisher Scientific, Waltham, MA, USA) for one hour, followed by washing. Bands were detected using enhanced chemiluminescence (Amersham Pharmacia Biotech, Piscataway, NJ, USA) and were imaged using an ImageQunat RT ECL machine coupled with Image Quanta TL 10.1-Capture software (https://us.vwr.com/assetsvc/asset/en_US/id/16613606/contents; accessed on 21 February 2024) (GE Healthcare Bio-Sciences AB, Bloomington, IL, USA).

### 4.2. Microbiological Experiments and Model-Based Prediction of Bacterial Growth Dynamics

#### 4.2.1. Bacterial Strains and Culture Conditions

*Staphylococcus aureus* ATCC 25923, *Escherichia coli* ATCC 53868, *Bacillus subtilis* ATCC 6633, *Bacillus cereus* ATCC 11778, and *Pseudomonas aeruginosa* ATCC 27853 were aseptically inoculated onto blood agar (Oxoid, Basingstoke, UK) plates and subsequent overnight incubation at 35 °C was employed to cultivate individual colonies. The purity of the bacterial cultures was maintained.

#### 4.2.2. Bacterial Growth Curve Measurements with a Microplate Reader and Colony-Forming Unit Assay

Bacterial cultures were meticulously prepared and bacterial density was adjusted to a concentration of 1 × 10^7^ colony-forming units per mL (CFU/mL). Subsequently, the standardised bacterial suspension was introduced into the wells of a microtiter plate (Thermo Fisher Scientific Inc., USA). The microtiter plate was then positioned within a Microtiter plate reader (Thermo Fisher Scientific Inc., USA), where, at regular 30 min intervals, the optical density (OD) of each well was measured at 620 nm [33]. This measurement involves the transmission of light through the well, with absorbance quantified accordingly. Importantly, to prevent biofilm formation, the microtiter plate was maintained in shaking mode between readings. The OD values serve as a quantitative indicator of bacterial growth, reflecting variations in light scattering and absorption as the bacteria proliferate. The microtiter plate reader systematically collects OD data at predetermined time points throughout a 24 h duration, forming a comprehensive time series of OD readings for each well, which are subsequently utilised to construct a detailed growth curve. Bacteria treated with phosphate-buffered solution (PBS) were considered as the negative control throughout the experiment; whereas, bacteria growing in MHB were considered as the positive growth control. Moreover, gentamicin 10 µg/mL served as the negative growth control, as previously described [34]. The percentage bacterial growth difference was determined with respect to negative control and results were expressed as mean  ±  standard error of the mean. All analyses were performed in three biological triplicates and each biological replicate consisted of three technical replicates. 

The bacteria that showed the highest growth inhibition when treated with milk EVs at 9 h of incubation in the preliminary screening study were selected for further study. Three aliquots of milk EVs at 50 µL each were introduced at distinct time points—0 h, 3 h, and 6 h from the initiation of the initial incubation. Consequently, four distinct test groups were maintained, as follows: Set 1, without any EV doses; Set 2, receiving the 1st dose solely at 0 h; Set 3, exposed to the 1st and 2nd doses at 0 h and 3 h; and Set 4, subjected to the 1st, 2nd, and 3rd doses at 0 h, 3 h, and 6 h, respectively, with observations extended until the completion of the 24 h [33]. At 30 min intervals, absorbance readings were systematically acquired, thereby facilitating the generation of a comprehensive temporal series of absorbance data for each test group. Furthermore, to prevent biofilm formation, the microtiter plate was maintained in shaking mode in between readings throughout the experiment.

The colony-forming unit (CFU) reduction assay was simultaneously conducted to assess bacterial growth dynamics at various time points (0 h, 3 h, 6 h, 12 h, 18 h, and 24 h). Bacterial cultures extracted from the above-mentioned co-culture of bacteria and EVs were subjected to further analysis. One microliter of aliquots of the incubating bacterial samples was aseptically spread onto blood agar plates using sterile spreaders. The spreading process ensured uniform distribution across the agar surface. At each designated time point, the spread plating procedure was repeated and the plates were incubated for colony formation. Following incubation, the number of colonies was counted. The obtained data were then employed to calculate the colony-forming units per millilitre (CFU/mL) for each time point, as indicated elsewhere [35] The experiment was conducted with three replicates.

#### 4.2.3. Model-Based Prediction of Time Derivatives of Bacterial Growth and Data Analysis

The R package “gcplyr” was used to calculate the lag times [30]. Growth curve metrics were calculated in a model-free manner by identifying features of the density data and its derivatives. Both plain and per-capita derivatives were calculated. Plain derivatives were calculated as the slope of the density data over time. Per-capita derivatives were calculated as the slope of the log-transformed density over time. “gcplyr” uses each pair of subsequent points to calculate derivatives. Lag time was defined as “the amount of time that passes between the start of a growth curve and the beginning of exponential growth” and was calculated as a projection of maximum growth rate fit back to the starting density. 

## 5. Conclusions

In conclusion, our study highlights the significant potential of milk-derived EVs as a promising avenue in combatting pathogenic bacteria, particularly *S. aureus*. The results unequivocally showcase the efficacy of these vesicles in inhibiting the growth of *S. aureus* ATCC 25923 in a dose-dependent manner. Notably, the observed extension of lag time and increased generation time further underscores the dynamic inhibition mechanism mediated by EVs. This aligns with our hypothesis, emphasising the inherent antimicrobial properties of milk EVs and their potential as alternative therapeutic interventions against drug-resistant infections. The consistent size and structural integrity of EVs highlight their suitability as valuable assets in addressing the escalating crisis of AMR worldwide. While recognising the study’s limitations, our findings propel the understanding of EVs as formidable agents in combatting AMR, offering a promising direction for future research and intervention strategies in the global battle against drug-resistant pathogens.

## Figures and Tables

**Figure 1 ijms-25-04759-f001:**
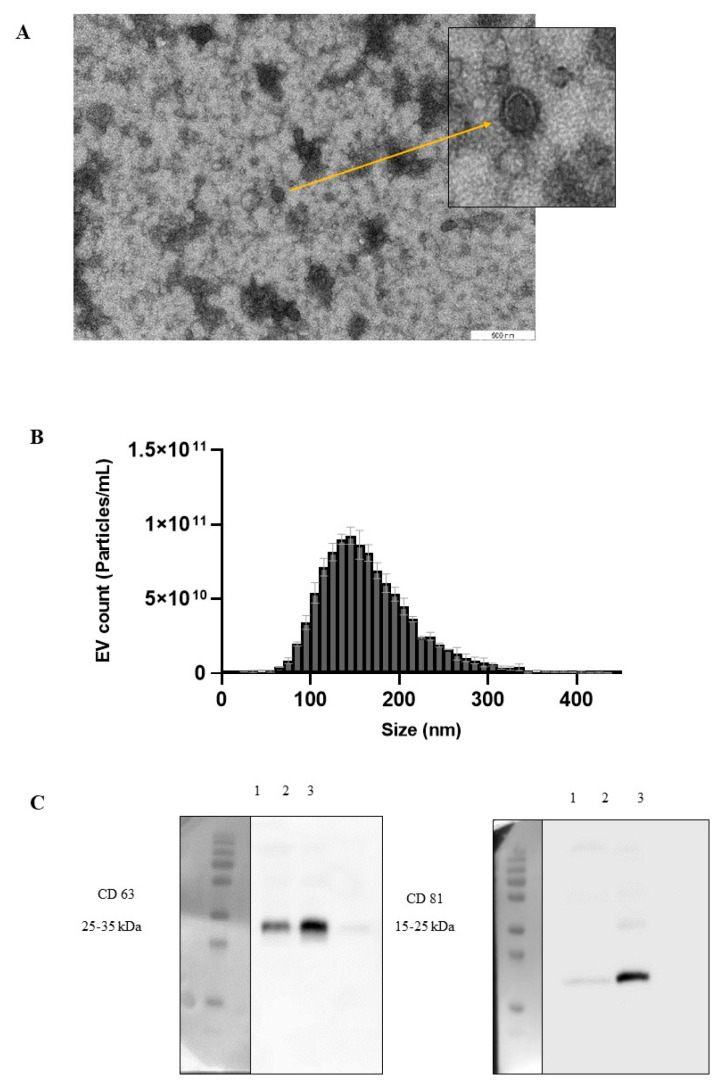
Characterisation of milk EVs. Morphology of milk EVs imaged using TEM. Scale bars are 1000 nm (**A**) and the particle size distribution and concentration of milk EVs are determined using ZetaView^®^ NTA. The particle size distribution of EVs is shown as particles/mL (mean ± SD, *n* = 3) (**B**). Western blot analysis confirmed the presence of CD63 and CD81 EV markers [1—−80 °C-stored TFF milk EVs; 2—lyophilised milk EVs; 3—whey (before TFF)] (**C**).

**Figure 2 ijms-25-04759-f002:**
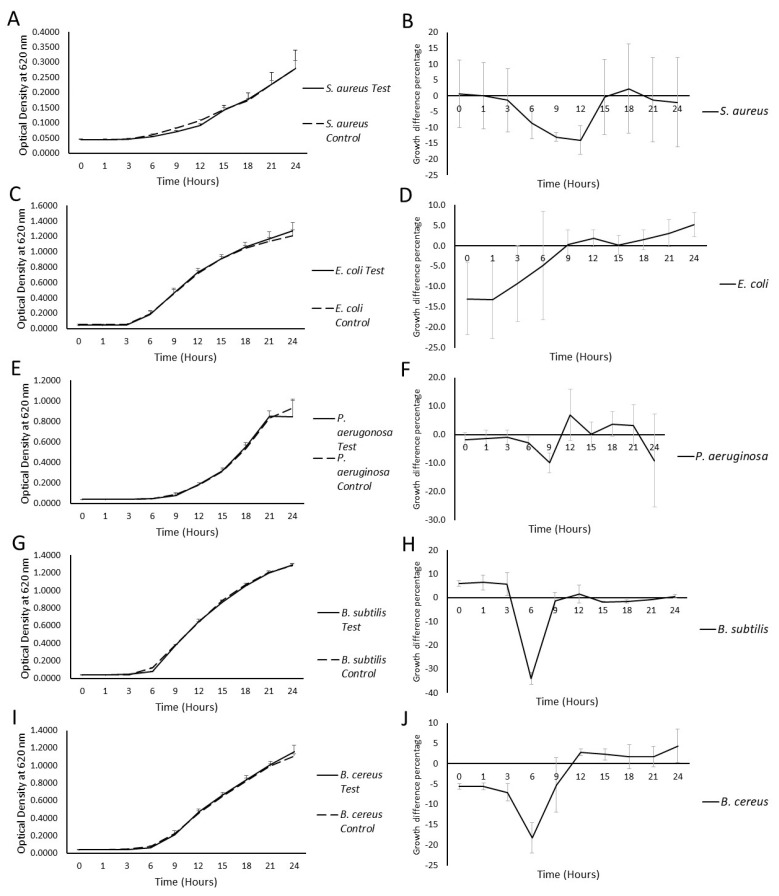
Microplate growth inhibition assays showing the activity of bacteria treated with milk EVs (**A**,**C**,**E**,**G**,**I**) and their relative growth inhibition (**B**,**D**,**F**,**H**,**J**).

**Figure 3 ijms-25-04759-f003:**
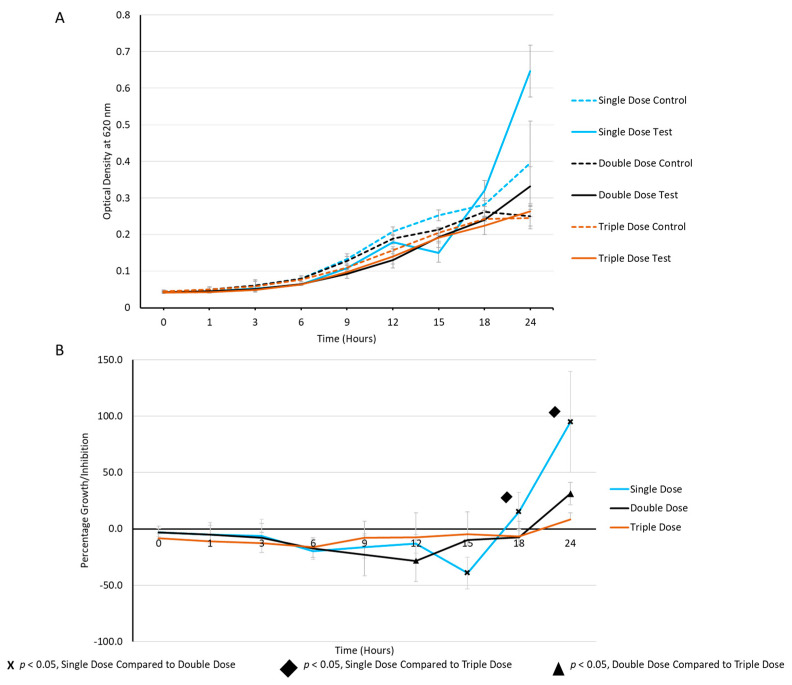
Dose-dependent inhibition was observed via OD values. Growth inhibition with a single dose, two doses, and three doses (**A**) and the relative percentage growth inhibition (**B**) of *S. aureus* ATCC 25923.

**Figure 4 ijms-25-04759-f004:**
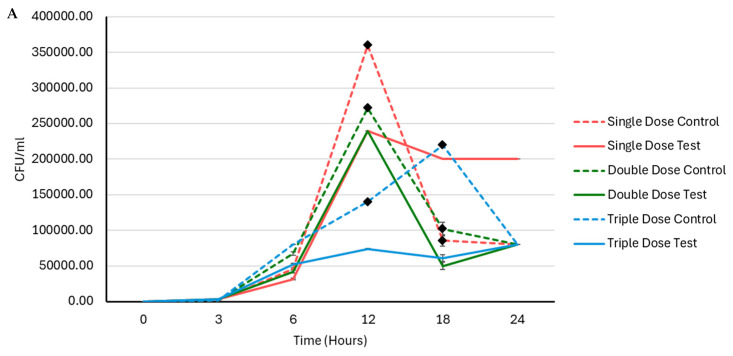
Dose-dependent inhibition observed via CFU reduction assay (**A**) and the relative percentage growth inhibition (**B**) of *S. aureus* ATCC 25923.

**Figure 5 ijms-25-04759-f005:**
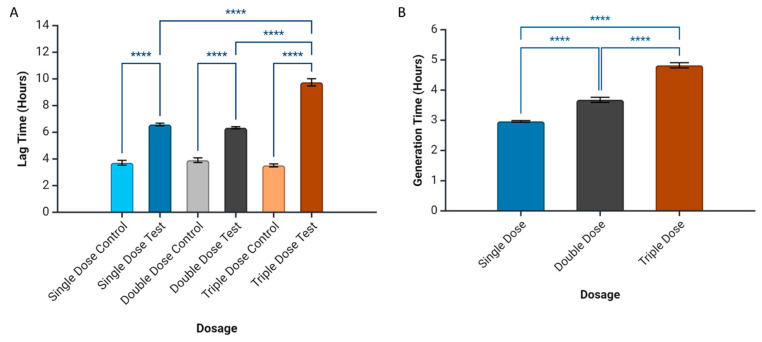
Model-based prediction of extension of lag phase (**A**) and effect on periodic dosing of milk EV towards increased generation time (**B**). **** *p* ≤ 0.05.

**Figure 6 ijms-25-04759-f006:**
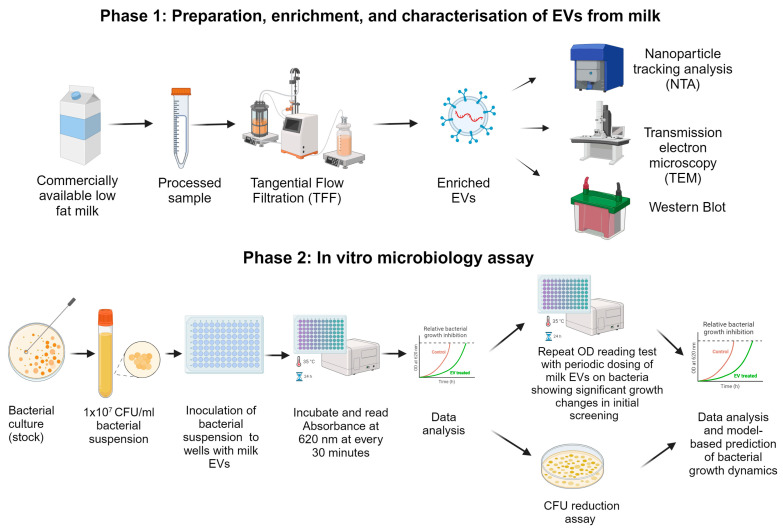
The experimental design for the preparation, enrichment, and characterisation of EVs from commercially available milk and the application of EVs in microbiological assays.

**Table 1 ijms-25-04759-t001:** Maximum percentage inhibition of bacteria treated with milk EVs at 9 h.

Bacterial Strain	Maximum Percentage Growth Difference (%, *p*-Value)					
	1 h	3 h	6 h	9 h	12 h	15 h
*Staphylococcus aureus* ATCC 25923	+0.10 ± 10.49, (0.8856)	−1.39 ± 9.96, (0.7397)	−8.44 ± 5.11, (0.2445)	−12.9 ± 1.39, (**0.0222**)	−13.96 ± 4.48, (**0.04506**)	−0.31 ± 11.9, (0.9454)
*Bacillus subtilis* ATCC 6633	+6.46 ± 3.10, (0.1484)	+5.69 ± 4.84, (0.2738)	−33.95 ± 2.42, (**0.0106**)	−1.23 ± 3.37, (0.6450)	+1.59 ± 3.81, (0.6739)	−1.80 ± 0.29, (0.3768)
*Bacillus cereus* ATCC 11778	−5.51 ± 0.82, (0.2983)	−7.05 ± 2.17, (0.2168)	−18.17 ± 3.72, (0.4195)	−5.21 ± 6.72, (0.4195)	+2.81 ± 0.79, (0.7679)	+2.29 ± 1.34, (0.7535)
*Escherichia coli* ATCC 53868	−13.3 ± 9.37, (0.2811)	−9.2 ± 9.39, (0.3730)	−4.8 ± 13.26, (0.9499)	+0.3 ± 3.68, (0.9585)	+1.9 ± 2.03, (0.8131)	+0.2 ± 2.41, (0.9611)
*Pseudomonas aeruginosa* ATCC 27853	−1.29 ± 2.81, (0.7469)	−1.01 ± 2.67, (0.8640)	−2.88 ± 13.26, (0.8283)	+9.90 ± 3.53, (0.7168)	+6.94 ± 9.09, (0.7452)	+0.16 ± 4.27, (0.9587)

## Data Availability

All the required data are available in the manuscript. Any other data that supports the findings of this study are available from the corresponding author upon reasonable request.

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
