# Peer review of "Harnessing Nature’s Defence: The Antimicrobial Efficacy of Pasteurised Cattle Milk-Derived Extracellular Vesicles on Staphylococcus aureus ATCC 25923"

_ijms, 2024, doi:10.3390/ijms25094759_

Round 1
Reviewer 1 Report
Comments and Suggestions for Authors
In the work „Harnessing Nature's Defence: The Antimicrobial Efficacy of 2 Pasteurized Cattle Milk-Derived Extracellular Vesicles on Staphylococcus aureus ATCC 25923“, the authors Dulmini Nanayakkara Sapugahawtte et al. present a nice and informative work. The paper is well written, however, some aspects might be improved.
Regarding resilient pathogens: better write resistant pathogens
The introduction and discussion contain some unnecessary repetitions and might thus be shortened; on the other hand, many inetersting aspects are not covered, like detailed information on the composition of milk vesicles, e.g. aquaporins, and biological effects that have so far been described.
The Western blot shown in fig. 1c does not contain a loading control.
In table 1, the authors show growth differences of bacteria compared to control at 9 hours. They should also show such numbers from other timepoints.
To me, the described effect looks rather small - what would the difference look like if other compounds, e.g. antibiotics, were used instead?
In table 4, the authors do not always show the error bars.
The authors used phosphate buffer solution (PBS) as negative control. However, they should also use vesicles of other source, since other effects of vesicles might be relevant, e.g. influening the optical density measurement unspecifically.
The authors did only use one source of milk. Could not milk from cattle of different origin and different raising or living conditions have completely different properties?
In the discussion, the authors should give some more ideas about which components oft he vesicles might be relevant fort eh effects observed.
Author Response
Responses to comments from the reviewer 1
The authors are very grateful to the reviewers for their valuable time and constructive comments which help improve the quality and clarity of our manuscript. We would like to address their comments point-by-point below.
Reviewer 1
Comments and Suggestions for Authors
In the work „Harnessing Nature's Defence: The Antimicrobial Efficacy of 2 Pasteurized Cattle Milk-Derived Extracellular Vesicles on Staphylococcus aureus ATCC 25923“, the authors Dulmini Nanayakkara Sapugahawtte et al. present a nice and informative work. The paper is well written, however, some aspects might be improved.
- Regarding resilient pathogens: better write resistant pathogens
Reply: Comment accepted by the authors and the word “resilient" has been replaced by the word “resistant” in lines 45 and line 186
- The introduction and discussion contain some unnecessary repetitions and might thus be shortened; on the other hand, many interesting aspects are not covered, like detailed information on the composition of milk vesicles, e.g. aquaporins, and biological effects that have so far been described.
Reply: The introduction has been modified and new insights have been incorporated in lines 58-80
- The Western blot shown in fig. 1c does not contain a loading control.
Reply: For the western blot image shown in Figure 1 C, loaded samples it into the gel by normalizing to a total protein concentration of samples. However, we did not run a loading control like β-Acin or GAPDH for this particular blot since our objective was mainly to identify the presence of the EVs in the enriched fractions based on known EV markers CD 63 and CD 81 only as described in previously published articles [1-3].
Reference:
- Go, G.; Jeon, J.; Lee, G.; Lee, J. H.; Lee, S. H. Bovine milk extracellular vesicles induce the proliferation and differentiation of osteoblasts and promote osteogenesis in rats. Journal of Food Biochemistry 2021, 45.
- Ukkola, J.; Pratiwi, F. W.; Kankaanpää, S.; Abdorahimzadeh, S.; KarzarJeddi, M.; Singh, P.; Zhyvolozhnyi, A.; Makieieva, O.; Viitala, S.; Samoylenko, A.; Häggman, H.; Vainio, S. J.; Elbuken, C.; Liimatainen, H. Enrichment of bovine milk-derived extracellular vesicles using surface-functionalized cellulose nanofibers. Carbohydrate Polymers 2022, 297, 120069.
- Wang, M.; Cai, M.; Zhu, X.; Nan, X.; Xiong, B.; Yang, L. Comparative proteomic analysis of milk-derived extracellular vesicles from dairy cows with clinical and subclinical mastitis. Animals 2023, 13, 171.
- In table 1, the authors show growth differences of bacteria compared to control at 9 hours. They should also show such numbers from other time points.
Reply: As requested by the reviewer, the authors have updated Table 1 incorporating bacterial growth difference at 1h, 3h, 6h, 9h, 12h and 15h (6 time points)
- To me, the described effect looks rather small - what would the difference look like if other compounds, e.g. antibiotics, were used instead?
Reply: Following previously published methodology, Gentamicin at a concentration of 10µg/ml served as the negative growth control, while phosphate-buffered saline (PBS) served as the negative control. Staphylococcus aureus cultured in Mueller-Hinton broth (MHB) was employed as the positive growth control. However, despite the anticipated susceptibility of the bacteria to Gentamicin, no growth was observed following 24 hours of incubation. This information has been incorporated into the methodology section, specifically in lines 384-386.
- In table 4, the authors do not always show the error bars.
Reply. Error bars are not always visible due to the extremely low variance among the biological replicates. However, they are added to the whole range of the graph.
- The authors used phosphate buffer solution (PBS) as negative control. However, they should also use vesicles of other source, since other effects of vesicles might be relevant, e.g. influening the optical density measurement unspecifically.
Reply: As depicted in Figure 1 below, we conducted experiments by placing media, PBS, and EVs in separate wells within the same 96-well plate. This strategy was implemented to ensure that the addition of EVs to bacteria as test samples and PBS as negative controls did not introduce unspecific influences on optical density readings. By maintaining this experimental setup, we aimed to accurately attribute any observed effects to the specific experimental conditions. Moreover, our methodology was carefully designed to align with established protocols and maintain consistency with prior studies [4].
Figure 1: S.aureus 10^7 CFU/ml growth in the presence of a single dose of 10^9 particles/ml milk EVs
Reference:
- Leiva-Sabadini, C.; Alvarez, S.; Barrera, N. P.; Schuh, C. M.; Aguayo, S. Antibacterial effect of honey-derived exosomes containing antimicrobial peptides against oral streptococci. International Journal of Nanomedicine 2021, Volume 16, 4891–4900.
8. The authors only use one source of milk. Could not milk from cattle of different origin and different raising or living conditions have completely different properties?
Reply: The reviewer raises an important point regarding the potential variability in milk properties based on the origin and living conditions of cattle. Indeed, milk composition can be influenced by factors such as breed, diet, environment, and management practices. However, our study focused on commercially available pasteurized milk and we never know the nature of the living conditions of cattle that milk has been collected. However, as a future direction of the study, exploring the effects of milk from diverse sources could provide valuable insights into the impact of these variables on the properties of milk-derived EVs
The raised issue is a future direction of the study and is incorporated in lines 323-325.
- In the discussion, the authors should give some more ideas about which components of the vesicles might be relevant fort the effects observed.
Reply: We acknowledge the reviewer's valuable suggestion regarding the exploration of potentially relevant components within EVs in forthcoming studies, given the current ambiguity regarding their cargo. Nonetheless, existing data have demonstrated the antimicrobial activity of certain antibacterial peptides found in bovine milk, such as Bovine k-casein, Kappacin A, and Isracidin, particularly against Gram-positive bacteria [5]. Consequently, it is plausible that these antimicrobial peptides could be enriched within EVs, thereby conferring visible antimicrobial effects. However, to address this aspect more rigorously, we intend to undertake transcriptomic analysis aimed at identifying crucial pathways that might contribute to the observed antimicrobial effects. This approach will provide valuable insights into the molecular mechanisms underlying the antimicrobial activity of EVs against pathogens. It's worth noting that this aspect has already been discussed in the manuscript within the "future directions" section, specifically in lines 270-272.
Reference:
- Singh, A.; Duche, R. T.; Wandhare, A. G.; Sian, J. K.; Singh, B. P.; Sihag, M. K.; Singh, K. S.; Sangwan, V.; Talan, S.; Panwar, H. Milk-derived antimicrobial peptides: Overview, applications, and future perspectives. Probiotics and Antimicrobial Proteins 2022, 15, 44–62.

Reviewer 2 Report
Comments and Suggestions for Authors
Manuscript entitled, “Harnessing Nature's Defence: The Antimicrobial Efficacy of Pasteurized Cattle Milk-Derived Extracellular Vesicles on Staphylococcus aureus ATCC 25923” is an interesting work to counter antimicrobial resistance by the pathogen with the help of Extracellular Vesicles (EV) from pasteurized cattle milk. The manuscript is well written and represented still I,’m having certain queries to be resolved prior to the publication of this work.
1. In abstract line 22-23 states “EVs from pasteurized cattle milk offer promise due to their unique properties.” The unique property of the EV should be clearly mentioned here to make it more significant.
2. In line 37-38 the data related of AMR is from the year 2019 which is quiet old and needs to be updated with more recent data.
3. In line 46-57 the authors have mentioned that Evs are also produced by the pathogens and can be helpful in their communication and transport of nutrients including proteins, lipids and nucleic acids but in the very next sentence authors are claiming it to be having antimicrobial character. This needs to be addressed by providing more insights on the mechanism of action involved in the antimicrobial effect of EV.
4. Full form of TFF should be used once in the manuscript for better understanding.
5. It has to be explained that the pasteurization process of milk is not having any impact on th EV for assuring that the antimicrobial effect is because of EV only.
6. Section “2.4. Prediction of bacterial growth-related parameters“, needs to be rephrased in detail for clear explanation of the applied methodology and obtained results.
7. Conclusion section also needs to be rephrased by emphasizing on obtained results and proposed hypothesis.
Author Response
Responses to comments from the reviewer 2
The authors are very grateful to the reviewers for their valuable time and constructive comments which help improve the quality and clarity of our manuscript. We want to address their comments point-by-point below.
Reviewer 2
Comments and Suggestions for Authors
Manuscript entitled, “Harnessing Nature's Defence: The Antimicrobial Efficacy of Pasteurized Cattle Milk-Derived Extracellular Vesicles on Staphylococcus aureus ATCC 25923” is an interesting work to counter antimicrobial resistance by the pathogen with the help of Extracellular Vesicles (EV) from pasteurized cattle milk. The manuscript is well written and represented still I,’m having certain queries to be resolved prior to the publication of this work.
- In abstract line 22-23 states “EVs from pasteurized cattle milk offer promise due to their unique properties.” The unique property of the EV should be clearly mentioned here to make it more significant.
Reply: This has been addressed in lines 58-66 “Initially regarded as cell membrane fragments for waste discharge, EVs have since been recognized for their involvement in various physiological processes, including angiogenesis, inflammation, immune response, and neuron signalling. Milk, notably rich in EVs, contains diverse RNA, lipids, and proteins within its EVs. The bilayer membrane structure of milk-derived EVs enables them to withstand gastric and pancreatic digestion, facilitating absorption by intestinal cells and subsequent biological activities. Furthermore, EVs, can traverse biological barriers and deliver cargo to target cells, make them attractive candidates for antimicrobial applications”
- In line 37-38 the data related of AMR is from the year 2019 which is quiet old and needs to be updated with more recent data.
Reply: Thank you for your feedback. We incorporated 2023 data into the manuscript [1].
Reference:
- WHO, 2023. Antimicrobial resistance https://www.who.int/news-room/fact-sheets/detail/antimicrobial-resistance#:~:text=It%20is%20estimated%20that%20bacterial,development%20of%20drug%2Dresistant%20pathogens. (accessed Apr 18, 2024).
- In line 46-57 the authors have mentioned that Evs are also produced by the pathogens and can be helpful in their communication and transport of nutrients including proteins, lipids and nucleic acids but in the very next sentence authors are claiming it to be having antimicrobial character. This needs to be addressed by providing more insights on the mechanism of action involved in the antimicrobial effect of EV.
Reply: Thank you for your insightful comment regarding the potential contradiction in our description of EVs both as facilitators of communication and nutrient transport, as well as possessing antimicrobial properties. We appreciate the opportunity to provide further clarification on this matter.
While EVs are indeed involved in intercellular communication and the transportation of various molecules, including proteins, lipids, and nucleic acids, recent research has also highlighted their role in host-pathogen interactions, particularly in the context of antimicrobial defence mechanisms.
The antimicrobial properties of EVs can be attributed to several factors:
- Cargo Delivery: EVs released by host cells can contain antimicrobial molecules such as antimicrobial peptides, microRNAs, and enzymes with antimicrobial activity. These cargo molecules can directly target and inhibit the growth or virulence of pathogens [2].
- Immune Modulation: EVs can modulate the immune response by delivering signalling molecules that activate innate immune pathways or promote the recruitment and activation of immune cells to combat microbial infections [3].
- Competition and Defense: EVs can compete with pathogen-derived vesicles for binding sites on host cells, thereby blocking the entry of pathogens or interfering with their ability to establish infection [4].
- Biofilm Disruption: EVs have been shown to disrupt microbial biofilms, which are protective structures formed by bacteria and other pathogens, making them more susceptible to antimicrobial agents and immune clearance [5].
- Quorum Sensing Regulation: Some EVs can interfere with microbial quorum sensing systems, which regulate virulence factor production and biofilm formation in bacterial populations, thereby attenuating pathogenicity [6].
To address your concern, we included a more detailed introductory paragraph of these mechanisms in our manuscript, emphasizing the multifaceted role of EVs in both promoting microbial communication and contributing to host defence against infections in lines 67-80.
Reference:
- Brakhage, A. A.; Zimmermann, A.-K.; Rivieccio, F.; Visser, C.; Blango, M. G. Host-derived extracellular vesicles for antimicrobial defense. microLife 2021, 2.
- Buzas, E. I. The roles of extracellular vesicles in the immune system. Nature Reviews Immunology 2022, 23, 236–250.
- Kuipers, M. E.; Hokke, C. H.; Smits, H. H.; Nolte-‘t Hoen, E. N. Pathogen-derived extracellular vesicle-associated molecules that affect the host immune system: An overview. Frontiers in Microbiology 2018, 9.
- Saad, M. G.; Beyenal, H.; Dong, W.-J. Dual roles of the conditional extracellular vesicles derived from pseudomonas aeruginosa biofilms: Promoting and inhibiting bacterial biofilm growth. Biofilm 2024, 7, 100183.
- Bose, S.; Aggarwal, S.; Singh, D. V.; Acharya, N. Extracellular vesicles: An emerging platform in gram-positive bacteria. Microbial Cell 2020, 7, 312–322.
- Full form of TFF should be used once in the manuscript for better understanding.
Reply: The full form of TFF was added in line 109
- It has to be explained that the pasteurization process of milk is not having any impact on th EV for assuring that the antimicrobial effect is because of EV only.
Reply: The study outlined in the provided source indicates that the pasteurization process applied to goat milk had minimal influence on the size, structure, and certain components of EVs. Therefore, it can be inferred that any antimicrobial effects observed are predominantly attributed to the EVs themselves rather than the pasteurization process [7].
Reference:
- Zhu, L.; Fu, S.; Li, L.; Liu, Y. Changes of extracellular vesicles in goat milk treated with different methods. LWT 2022, 170, 114038.
- Section “2.4. Prediction of bacterial growth-related parameters“, needs to be rephrased in detail for clear explanation of the applied methodology and obtained results.
Reply: The requested change has been done in the manuscript under section 2.4.
- Conclusion section also needs to be rephrased by emphasizing on obtained results and proposed hypothesis.
Reply: The conclusion was rewritten as below in lines 453-464
“In conclusion, our study highlights the significant potential of milk-derived EVs as a promising avenue in combating pathogenic bacteria, particularly S. aureus. The results unequivocally showcase the efficacy of these vesicles in inhibiting the growth of S. aureus ATCC 25923 in a dose-dependent manner. Notably, the observed extension of lag time and increased generation time further underscores the dynamic inhibition mechanism mediated by EVs. This aligns with our hypothesis, emphasizing the inherent antimicrobial properties of milk EVs and their potential as alternative therapeutic interventions against drug-resistant infections. The consistent size and structural integrity of EVs highlight their suitability as valuable assets in addressing the escalating crisis of AMR worldwide. While recognizing the study's limitations, our findings propel the understanding of EVs as formidable agents in combating AMR, offering a promising direction for future research and intervention strategies in the global battle against drug-resistant pathogens.”

Round 2
Reviewer 1 Report
Comments and Suggestions for Authors
My concerns have been met.
Reviewer 2 Report
Comments and Suggestions for Authors
The authors have resolved all the queries raised by the reviewers and can be accepted for publication in its current form